# Diffusion Language Model with Query-Document Relevance for Query-Focused Summarization

**Shaoyao Huang, Luozheng Qin** and **Ziqiang Cao**[*]
Institute of Artificial Intelligence, Soochow University

{20224227022, 20225227060}@stu.suda.edu.cn, zqcao@suda.edu.cn

## Abstract

Query-Focused Summarization (QFS) aims to generate summaries from source documents that can answer specific queries. Although the QFS task has gained increasing attention recently, its development is constrained by the fact that mainstream QFS models are BART variants, which are autoregressive and suffer from long-term dependencies and exposure bias. To address these problems, we adopt a diffusion language model that performs well in non-autoregressive scenarios to effectively resolve issues related to autoregressive methods. However, QFS requires guidance from queries to generate adequate summaries, while diffusion language models have limited sensitivity to queries. In this paper, we propose QFS-DLM, a non-autoregressive diffusion language model that incorporates query-document fragment relevance and query-document global relevance to enhance the adaptability of QFS tasks. Firstly, we extract key fragments from documents based on queries and assign higher weights to them, thereby emphasizing crucial and continuous information within the document. Secondly, we calculate global relevance scores between queries and documents, and then integrate these scores into the model's loss function, enabling the model to prefer high-quality data and distance itself from low-quality data. Overall, our method achieves state-of-the-art performance on Debatepedia and Pub-MedQA datasets in ROUGE scores, GPT-4, and human evaluations.

## 1 Introduction

Query-Focused Summarization (QFS) aims to generate summaries that address specific queries by extracting crucial information from source documents (Dang, 2005). Inspired by the significant generation capabilities of BART (Lewis et al., 2019), researchers manage to adopt it into QFS

---

[*]Corresponding Author.

| *High Query-Document Relevance Score Sample* |
|---|
| **Query**: do gasoline economy standards help reducing emissions and combat global warming? |
| **Document**: the most basics function of fuel economy standards is that they help the average cars burn less gasoline so emit less # into the atmosphere. the net effect is a reduction in greenhouse gas emission into the atmosphere which lowers the net human contribution to global warming. |
| **Summary**: fuel economy requirements reduce emissions fight globally warming. |

| *Low Query-Document Relevance Score Sample* |
|---|
| **Query**: does meeting unfriendly nations helping diplomacy? |
| **Document**: michigan democrat congressman dave hoekstra responding to the notion of these requests predicted " that would be an unsupportable positions for the president of the united states to be putting in [ # ]. " |
| **Summary**: obama closed talks offer rogue leaders disinformation opportunities. |

Table 1: Examples are taken from the training set of Debatepedia. In the high relevance score sample, red represents the common keywords shared among the query, document, and summary. In the low relevance score sample, we only find the words president and obama, which have a slight connection.

to generate query-focused summary in an autoregressive(AR) manner (Su et al., 2021; Laskar et al., 2022; Park and Ko, 2022). However, traditional AR models used in QFS have limitations such as long-term dependencies (Bengio et al., 1994) and exposure bias (Bengio et al., 2015).

Recently, the diffusion language model GE-NIE (Lin et al., 2023) has achieved remarkable performance without operating AR. In the case of non-autoregressive (NAR) modeling, GENIE significantly outperforms BART. This performance gap is

because BART operates fully-NAR, while GENIE adopts an iter-NAR approach. Huang et al. (2022) shows a gap called conditional total correlation between AR and fully-NAR learning paradigms because of the lossy decomposition of NAR models. However, when comparing iter-NAR with AR models, they both can be factorized into an initial prediction term and a progressive prediction process based on different contexts (i.e., left-context in AR and full-context in iter-NAR) (Gong et al., 2022), and the discrepancy pointed out by Huang et al. (2022) is therefore closed in iter-NAR assuming sufficient steps. By showing GENIE is an extension of the iter-NAR model, we offer a justification that it will not suffer from the conditional total correlation for the same reason.

Despite these advantages, two problems hinder the diffusion language model from achieving optimal performance when applied to the QFS task. Firstly, different fragments in the document have varying degrees of impact on the summary, and treating them equally does not reflect the specificity of query-guided summary generation. Secondly, the query-document global relevance indirectly reflects the data quality (see Table 1), and low-quality data can significantly impair the model's performance.

To address the first problem, we referred to the work of QFS-BART(Su et al., 2021), QFS-BART method utilizes a QA model to generate the answer relevance score for each word in the document and adds this relevance score to the cross-attention. We believe that it is more effective to average the scores for the document fragment that the QA model identifies as most relevant to the query, as the document fragment recognized by the QA model has a high degree of correlation with the summary. To address the second problem, we calculate the query-document global relevance score and assign higher/lower loss weights to high-quality/low-quality data, enhancing the positive impact of high-quality data and reducing the impairment of low-quality data.

In this study, we propose QFS-DLM [1], a diffusion language model that incorporates query-document relevance to generate high-quality summaries. Firstly, we incorporate the query-document fragment relevance to make the model pay more attention to continuous information within a key

fragment of the document. Secondly, we incorporate the query-document global relevance into the model's loss function, enabling it to prefer high-quality data and distance itself from low-quality data. Finally, to validate the effectiveness of our proposed approach, we conduct extensive experiments on popular QFS datasets, including Debatepedia (Nema et al., 2017) and PubMedQA (Jin et al., 2019).

Our contributions are as follows:

- We incorporate query-document fragment relevance into the diffusion language model, assigning higher weights to the document fragment that the QA model identifies as most relevant to the query, enabling the model to focus more on the continuous information in the key fragment.

- We incorporate the query-document global relevance into the model's loss function, enabling it to prefer high-quality data and distance itself from low-quality data.

- We conduct experiments on the Debatepedia and PubMedQA datasets, achieving state-of-the-art performance in terms of ROUGE scores. Furthermore, our approach demonstrates promising results in the GPT-4 and human evaluations.

## 2 Related Work

### 2.1 Query-focused Summarization

In the QFS task, evaluating query relevance is critical for generating a summary focused on the query. Early works (Lin et al., 2010; Shen and Li, 2011) focused on extracting query-related sentences as summaries, while Li and Li (2014) improved this procedure by compressing the extracted sentences.Nema et al. (2017); Hasselqvist et al. (2017) proposed neural abstraction models with additional query attention mechanisms to generate query-focused summaries. Deng et al. (2020a) treated the relationship between query and source sentences as a multi-hop reasoning process and generated summaries by integrating information from different reasoning steps. Additionally, researchers utilized question-answer models to find potential query-related evidence in QFS. Xu and Lapata (2020) employed question-answer models for ranking sentence-level or paragraph-level answer evidence. Su et al. (2021) incorporated answer

---

[1]The code is available at: `https://github.com/ShaoyaoHuang/QFS-DLM/tree/main`

| Symbols | Description |
|---|---|
| $F$ | fragment relevance score sequence |
| $t$ | time step |
| $\beta_t$ | $\beta_t \in (0, 1)$ as different variance scales |
| $x_t$ | continuous latent representation of the gold summary |
| $H_S$ | source text representation. |
| $\alpha_t$ | $\alpha_t = 1 - \beta_t$ |
| $R_i$ | global relevance score of the $i$-th sample between the query and the document |
| $Emb(y)$ | embedding of the gold summary |
| $u_\theta^{t-1}$ | predicted noise |
| $\hat{\mu}_{t-1}$ | actual noise |
| $z_\theta$ | output of the denoising architecture |
| $I$ | covariance matrix of Gaussian distribution |

Table 2: Parameter meaning explanation.

relevance scores generated by a question-answer model as explicit fine-grained query relevance into a transformer-based abstract summarization model. Yuan et al. (2022) integrated task knowledge from text summarization and question answering into a well-designed prefix and applied it. Park and Ko (2022) incorporated a graph attention mechanism that calculated the relevance of word nodes to query nodes. Overall, query-document relevance is crucial for query-focused summarization, and we incorporate it into our model.

## 2.2 Diffusion Language Models

With the great success of diffusion models in computer vision, some researchers have started exploring the application of diffusion models in text generation. Hoogeboom et al. (2021) introduced polynomial diffusion for character-level text generation, applying forward classification noise with a Markov transition matrix. Austin et al. (2021) generalized discrete text diffusion models by introducing an absorbing state [MASK]. However, discrete diffusion models may be influenced by the scale of one-hot row vectors and only generate text samples unconditionally in the discrete space. Li et al. (2022) proposed a novel continuous latent representation language model with different mapping functions that connect text's discrete and continuous spaces. Gong et al. (2022) employed an end-to-end, classifier-free approach to guide the diffusion conditional generation and provided a comparative analysis between the diffusion, autoregressive, and non-autoregressive models. Lin et al. (2023) proposed a large-scale pre-trained diffusion language model, which can generate high-quality texts for sequence-to-sequence tasks. Compared to previous works, we focus on improving the adaptability of the diffusion language model for the QFS task.

## 3 Methodology

In this section, we present our approach to incorporating query-document relevance into the diffusion language model. First, we describe the method of incorporating query-document fragment relevance. Then, we introduce the query-document global relevance, which is only used during the training phase, and our model architecture as shown in Figure 1.

Based on the performance of the diffusion language model GENIE [2] (Lin et al., 2022) in the general text summarization, we choose it as our base model. The GENIE has undergone extensive pretraining and adopts the sequence-to-sequence framework. Unlike the traditional autoregressive text generation paradigm that generates one token at a time, GENIE parallelly outputs embedding sequences at each denoising step, making it a non-autoregressive model. our parameters is presented Table 2.

## 3.1 Query-Document Fragment Relevance

In recent years, neural models (Yang et al., 2019a; Su et al., 2019) have shown remarkable achievements in QA tasks. In order to apply QA models to the QFS task, we use HLTC-MRQA (Su et al., 2019) to generate the answer relevance score for each word in context. The reason for choosing HLTC-MRQA is twofold: 1) it shows robust generalization and transferring ability on different datasets, and 2) the model shows great performance in QA tasks and significantly outperforms the BERT-large baseline by a large margin. The HLTC-MRQA is introduced as follows.

Based on XLNet (Yang et al., 2019b), HLTC-MRQA is fine-tuned on multiple QA datasets. Given a context that contains $n$ words, the model

---

[2] https://github.com/microsoft/ProphetNet/tree/master/GENIE

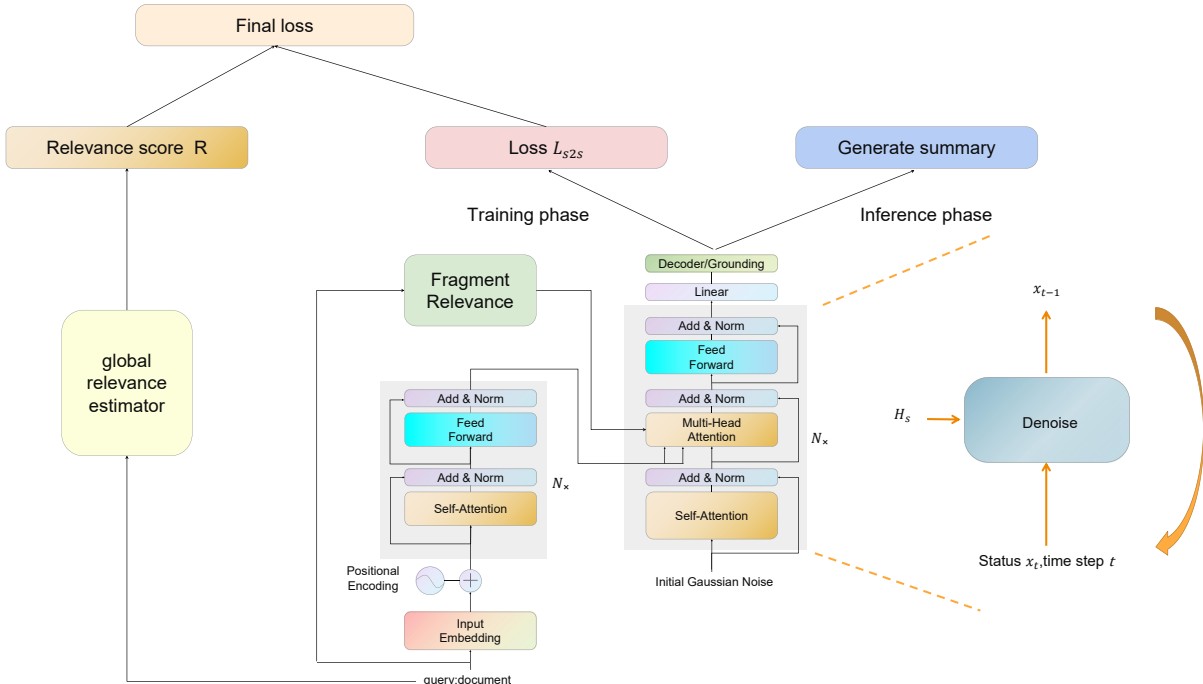

Figure 1: The framework of QFS-DLM. For query-document fragment relevance, we use a QA model to extract a key fragment from the document and add weight to it in cross-attention. During the training process, we calculate the query-document global relevance scores and incorporate them into the model's loss function.

outputs a distribution $s \in (0, 1)$ for each word's probability of being the start word of the answer and a probability distribution $e \in (0, 1)$ for being the end word of the answer. Su et al. (2021) approach involves adding the scores of $s$ and $e$ and incorporating this score into the hidden state of the corresponding words in the model's cross-attention. However, this approach can disrupt the continuity information of important fragments and the original task of the QA model is to separately identify the highest scores for $s$ and $e$, considering the middle part between $s$ and $e$ as the answer (if the position of $s$ is after $e$, then it is considered that there is no fragment to answer the query). To address these two issues, we introduce an additional weighting mechanism specifically for the fragments between $s$ and $e$. To achieve this, we generate a relevance score sequence $F$, represented as follows:

$$F = [1, 1, ..., 1, 2^*, 2, ..., 2^{\#}, 1, 1, ..., 1] \quad (1)$$

where $*$ represents the position corresponding to the highest score for $s$, and $\#$ represents the position corresponding to the highest score for $e$.

Next, we incorporate this relevance score into the model. The encoder and decoder of the diffusion language model are both composed of

6 layers of Transformers. The decoder stably generates high-quality text iteratively, and cross-attention (Vaswani et al., 2017) connections are established between the encoder and decoder. To enable the decoder to better focus on crucial and contiguous information within the key fragments of the document, we increase the weighting of key fragments in the cross-attention mechanism. The original formula for cross-attention is as follows:

$$Attention(Q, K, V) = softmax(\frac{QK^T}{\sqrt{d}})V, \quad (2)$$

where the summary is input into the decoder to obtain the representation $Q$, where $d$ represents the dimension of $Q$. In order to enable interaction between the query and document information in the encoder, we concatenate the query and document as input to the encoder to obtain $K$ and $V$. Then, in our method, we incorporate the relevance score $F$ into the cross-attention, as follows:

$$Attention(Q, K, V) = softmax(\frac{Q(K * F)^T}{\sqrt{d}})V \quad (3)$$

By increasing the weights, the contiguous fragments corresponding to the query have a more significant impact on the generation of the summary

## 3.2 Query-Document Global Relevance

To apply query-document global relevance to the QFS task, we use Dense Passage Retrieval (DPR) (Karpukhin et al., 2020) to generate query-document relevance scores. There are two reasons for choosing DPR: (1) it demonstrates robust generalization and transferability across different datasets, and (2) the model performs well in question-answer tasks and is widely used on Hugging Face (with 160k downloads). DPR can effectively retrieve the most relevant documents from a large corpus in relation to a search query. We use the DPR[3] model to calculate the relevance score between the query and the document. This is accomplished through the following equation:

$$DPR_{Sim}(D,Q) = \frac{\vec{D} \cdot \vec{Q}}{||\vec{D}|| \cdot ||\vec{Q}||} \quad (4)$$

where $\vec{D}$ is the embedding vector of the input document, and $\vec{Q}$ is the embedding vector of the input query.

For direct interaction between the query and document within the bidirectional self-attention, we format the source text as "$[CLS]\ document\ [SEP]\ query$". The encoder computes representations for all tokens and outputs hidden vectors, enabling the source text to be represented as $H_s$:

$$H_s = Encoder([CLS]\ document\ [SEP]\ query) \quad (5)$$

where the encoder refers to a 6-layer transformer.

**Training Phase** To train the diffusion language model for the QFS task, we first need to transform the gold summary $y$ into a continuous state $x_0$. We achieve this by applying the embedding function and perturbing it with Gaussian noise. This process can be represented by the equation:

$$q(x_0|y) = \mathcal{N}(x_0; Emb(y), \beta_0\mathbf{I}) \quad (6)$$

where $Emb(y)$ represents the embedding of the gold summary, and $\beta_0$ represents the scaling of the variance at time step $t = 0$.

Given a gold summary distribution $x_0$, gradually adds Gaussian noises to $x_0$ according to a variance scheduler $\beta_1, ..., \beta_T$, corrupting x0 into a standard Gaussian noise $x_t \sim \mathcal{N}(0, \mathbf{I})$. At the time step

[3]https://huggingface.co/facebook/dpr-ctx_encoder-multiset-base

$t + 1$, the latent variable $x_{t+1}$ is only determined by the $x_t$ at time $t$, expressed as:

$$q(x_{t+1} \mid x_t) = \mathcal{N}(x_{t+1}; \sqrt{1 - \beta_{t+1}}x_t, \beta_{t+1}\mathbf{I}) \quad (7)$$

where $\beta_{t+1} \in (0,1)$, as $t$ increases, $x_t$ becomes closer to standard Gaussian noise $\mathcal{N}(x_t; 0, \mathbf{I})$.

We then apply the forward diffusion process (equation 7) to obtain the state $x_t$ at step $t$ as a function of $x_0$, using the following equation:

$$q(x_t|x_0) = \mathcal{N}\left(x_t; \sqrt{\bar{\alpha}_t}x_0, \sqrt{1 - \bar{\alpha}_t}\mathbf{I}\right) \quad (8)$$

where $\alpha_t = 1 - \beta_t$, and $\bar{\alpha}_t = \prod_{i=1}^{t} \alpha_i$.

During the training phase, we randomly select a step $t$ to compute $x_t$ and then utilize a denoising architecture that predicts noise through cross-attention with the source text representation $H_S$. The equation below provides the predicted noise:

$$u_\theta^{t-1} = \frac{1}{\sqrt{\alpha_t}}(x_t - \frac{\beta_t}{\sqrt{1 - \bar{\alpha}_t}}z_\theta(x_t, t, H_s)) \quad (9)$$

where $z_\theta$ represents the output of the denoising architecture and $\theta$ represents its parameters.

The training objective is to minimize the squared error between the predicted noise $u_\theta^{t-1}$ and actual noise $\hat{\mu}_{t-1}$, as well as the reconstruction error between $x_0$ and the gold summary embedding, as shown in the equation:

$$\mathcal{L}_{s2s} = \mathbb{E}_{q(x_{0:T}|y)}[\sum_{t=1}^{T} \left\| \mu_\theta^{t-1} - \hat{\mu}_{t-1} \right\|^2 \quad (10)$$
$$+ \left\| Emb(y) - \mu_\theta^0 \right\|^2 - \log p_\theta(y|x_0)]$$

where $\log p_\theta(y|x_0)$ represents mapping the continuous latent variable $x_0$ into the discrete space token of gold summary $y$.

Finally, we incorporate query-document global relevance as shown in the equation:

$$\mathcal{L}_{Final} = \frac{\sum_{i=1}^{B} e^{R_i} \cdot \mathcal{L}_{s2s}^i}{\sum_{i=1}^{B} e^{R_i}} \quad (11)$$

where $B$ represents the number of samples in a batch, $R_i$ denotes the relevance score of the $i$-th sample between the query and the document, and $\mathcal{L}_{s2s}^i$ represents the loss of the $i$-th sample in the diffusion language model.

**Inference Phase** To generate summaries, we start from the last step $t = T$, and sample a state $x_T$ from a standard Gaussian distribution. Then, we perform denoising, iteratively generate noise from the previous step, and subtract that noise from

| Models | Debatepedia | | | PubMedQA | | |
|---|---|---|---|---|---|---|
| | ROUGE-1 | ROUGE-2 | ROUGE-L | ROUGE-1 | ROUGE-2 | ROUGE-L |
| *Original results* | | | | | | |
| ChatGPT one-shot | 22.7 | 6.68 | 18.8 | 34.8 | 12.5 | 23.7 |
| SD2[†] | 41.3 | 18.8 | 40.4 | 32.3 | 10.5 | 26.0 |
| QR-BERTSUM-TL[†] | 58.0 | 45.2 | 57.1 | - | - | - |
| MSG[†] | - | - | - | 37.2 | 14.8 | 30.2 |
| QFS-BART[†] | 59.0 | 44.6 | 57.4 | 38.3 | 16.4 | 29.1 |
| PreQFAS[†] | 59.3 | 45.6 | 58.2 | - | - | - |
| QSG BART[†] | 64.9 | 52.3 | 63.3 | 38.4 | 17.0 | 29.8 |
| *Reproduction on GENIE* | | | | | | |
| QSG BART | 53.7 | 33.9 | 51.8 | 34.1 | 11.6 | 25.0 |
| QUERYSUM | 65.1 | 51.4 | 63.0 | 44.9 | 18.7 | 33.4 |
| Prefix-merging | 65.4 | 51.5 | 63.8 | 45.3 | 19.4 | 33.2 |
| QFS-BART | 65.9 | 51.8 | 64.2 | 45.5 | 19.6 | 33.0 |
| GENIE zero-shot | 29.8 | 9.7 | 24.9 | 29.4 | 7.3 | 20.1 |
| GENIE finetune | 65.4 | 51.7 | 63.9 | 45.2 | 19.6 | 33.6 |
| QFS-DLM fragment relevance | 66.5 | 52.4 | 64.9 | 45.6 | 20.1 | 33.9 |
| QFS-DLM global relevance | 66.2 | 52.5 | 64.8 | 46.3 | 20.9 | 34.8 |
| **QFS-DLM** | **67.3** | **53.1** | **65.7** | **46.5** | **21.0** | **35.1** |

Table 3: ROUGE-F1 scores for Debatepedia and PubMedQA datasets. QFS-DLM fragment/global relevance indicates that we only use fragment/global relevance separately, while QFS-DLM indicates that we use two types of relevance simultaneously. [†] denotes results from the original paper.

| Datasets | Summary Sources | Rank | | | |
|---|---|---|---|---|---|
| | | **fluency** | **faithfulness** | **coverage** | **overall** |
| Debatepedia | QFS-BART | 1 | 3 | 3 | 2 |
| | GENIE | 3 | 2 | 2 | 3 |
| | QFS-DLM | 2 | 1 | 1 | 1 |
| PubMedQA | QFS-BART | 2 | 3 | 3 | 3 |
| | GENIE | 3 | 2 | 2 | 2 |
| | QFS-DLM | 1 | 1 | 1 | 1 |

Table 4: The evaluation of summary quality using GPT-4. We feed the query, document, and shuffled summaries as inputs to GPT-4 to determine which summary performs the best. Subsequently, we rank the number of summaries assessed as the best for each specific metric.

| | Debatepedia | | | PubMedQA | | |
|---|---|---|---|---|---|---|
| | train | valid | test | train | valid | test |
| Samples | 12K | 0.7K | 1K | 168K | 21K | 21K |
| Avg Len Query | 11.6 | 11.7 | 11.3 | 15.3 | 15.4 | 15.3 |
| Avg Len Doc | 72.9 | 71.6 | 69.9 | 200 | 200 | 200 |
| Avg Len Sum | 9.9 | 9.8 | 9.9 | 37.6 | 37.6 | 37.5 |

Table 5: Statistics of datasets.

the current state to obtain $x_{t-1}$. Upon reaching $t = 0$, we apply the clamping trick (Li et al., 2022) to replace the value of $x_0$ with its nearest word embedding and then decode discrete tokens from $x_0$.

## 4 Experiment

### 4.1 Datasets

We evaluate our proposed model on two QFS datasets: Debatepedia (Nema et al., 2017) and Pub-MedQA (Jin et al., 2019). Debatepedia contains 12.7K samples crawled from 663 debates of 53 diverse categories in an encyclopedia of debates.

PubMedQA is a biomedical abstractive question-answer dataset that contains 210K samples. We follow the standard splits used in the original paper's methods. Table 5 presents the statistical information of these datasets.

### 4.2 Training Details

Our setup is largely consistent with GENIE (Lin et al., 2022). Specifically, we set the latent variable dimension to 768 and the embedding variable dimension to 128. During training, we utilize the Adam optimizer (Kingma and Ba, 2015) with a learning rate of 1e-4 and set the batch size to 64. We train our model using four 24G A5000 GPUs.

| **Input:** |
|---|
| **Document**: republicans argued in a june 2009 letter to president obama : " [ public insurance ] would be a federal government takeover of our healthcare system taking decisions out of the hands of doctors and patients and placing them in the hands of a washington bureaucracy . **Query**: is it important to give citizens the choice of a public insurance option ? From an **overall** perspective, which summary is best? If the first one is the best, output "1"; if the second one is the best, output "2"; if the third one is the best, output "3": **Summary 1**: public insurance would amounts to a " federal takeover " of health care **Summary 2**: public insurance takes health care decisions out of the hands of patients. **Summary 3**: public insurance could give citizens out of health bureaucracy. |
| **Output:** |
| **2** |

Table 6: An example of GPT-4 evaluation. **Summary 1** refers to QFS-BART summary, **Summary 2** is QFS-DLM summary, and **Summary 3** is GENIE summary. QFS-DLM summary is smoother than QFS-BART and can correct errors made by GENIE.

## 4.3 Comparative Methods

In this work, we compare QFS-DLM with several other models. These models include: (1) Chat-GPT One-Shot, which utilizes ChatGPT to generate summaries based on a provided example. (2) SD2 (Nema et al., 2017), which incorporates a query attention model and a diversity-based attention model in the encoding-attention-decoding paradigm. (3) QR-BERTSUM-TL (Laskar et al., 2020), which presents a query-relevance technique with the Transformer-based BERTSUM model. (4) MSG (Deng et al., 2020b), which utilizes the relevance between the query and the sentences and the mutual relationships computed by the attention mechanism. (5) QFS-BART (Su et al., 2021), which incorporates document and answer relevance. (6) PreQFAS (Laskar et al., 2022), which uses the CNN/DM dataset for data augmentation. (7) QSG BART (Park and Ko, 2022), which incorporates a graph attention mechanism that calculates the relevance of word nodes to query nodes. (8) GE-NIE (Lin et al., 2023), a large-scale pretrained diffusion language model.

## 4.4 Evaluation Metrics

We use ROUGE metrics (Lin, 2004), which include Rouge-1, Rouge-2, and Rouge-L, as evaluation metrics.

Additionally, we employ GPT-4 (Ouyang et al., 2022) and humans to assess the quality of the summaries based on four aspects: fluency, faithfulness, coverage, and overall performance. Prompt template examples of GPT-4 evaluation are shown in Table 6, where the order of the summaries is generated randomly. When using other metrics, the terms **overall** can be replaced. For the human evaluation of summaries quality, in order to ensure objectivity, we shuffled the summaries and asked five text summarization researchers to score the summaries. Each perspective was scored on a scale of 1 to 5 points, with a higher score indicating better performance.

## 4.5 Results

**ROUGE Metrics**: The experimental results for Debatepedia and PubMedQA are presented in Table 3, with the highest scores highlighted in bold. Our proposed model achieves significant improvements compared to previous works. Although our model outperforms the comparison models in all metrics on both datasets, it is evident that the performance on PubMedQA is better than that on Debatepedia. This may be because PubMedQA has much longer summaries (as shown in Table 5), and the non-autoregressive diffusion language model can alleviate the constraints of gold summary length.

We analyzed each method that reproduction on the GENIE separately: (1) QSG BART's reproduction yielded poor results, as its auxiliary task significantly disrupted the encoder of the diffusion language model. (2) QUERYSUM (Xu and Lapata, 2020) slightly reduced the effectiveness of the diffusion language model because sentence ordering in single-document datasets is not as critical as in multi-document datasets, and reordering can disrupt continuity between sentences. (3) Prefix-merging (Yuan et al., 2022) had minimal impact on the diffusion language model since it is designed for few-shot scenarios. (4) QFS-BART improved performance by focusing on keywords. Our fragment-relevance method better emphasized contiguous information, resulting in more noticeable performance gains. Overall, our approach exhibits greater adaptability and performs better on the QFS task than other methods.

| Datasets | summary sources | query-focusing | fluency | faithfulness | coverage | overall |
|---|---|---|---|---|---|---|
| Debatepedia | QFS-BART | 3.52 | 4.23 | 3.50 | 3.67 | 3.59 |
| | GENIE | 3.41 | 4.14 | 3.56 | 3.78 | 3.71 |
| | QFS-DLM | 4.07 | 4.53 | 4.09 | 4.16 | 4.13 |
| PubMedQA | QFS-BART | 3.12 | 4.03 | 3.27 | 3.24 | 3.28 |
| | GENIE | 2.95 | 3.85 | 3.46 | 3.47 | 3.54 |
| | QFS-DLM | 3.68 | 4.21 | 3.94 | 4.01 | 3.87 |

Table 7: Human evaluation on Debatepedia and PubMedQA datasets.

| Datasets | query-document relevance | high quality | low quality |
|---|---|---|---|
| Debatepedia | high relevance | 89.3% | 10.7% |
| | low relevance | 37% | 63% |
| PubMedQA | high relevance | 92.3% | 7.7% |
| | low relevance | 53.7% | 46.3% |

Table 8: Correlation between query-document global relevance and sample quality.

| |
|---|
| **Query**: Are zoos a poor means to protecting endangered species? |
| **Document**: Visitors to zoos may raise their awareness of endangered species by being directly exposed to them. |
| **ChatGPT Summary**: Zoos can actually be a valuable means of protecting endangered species. By allowing visitors to directly observe endangered species, zoos raise awareness and foster a connection between people and these animals. This can lead to increased support for conservation efforts and funding for protecting these species in the wild. Overall, zoos can play a positive role in educating the public and contributing to the conservation of endangered species. |
| **QFS-BART Summary**: Zoos provide children with an exposure to endangered species. |
| **QFS-DLM Summary**: Zoos help securing endangered species. |
| **Gold Summary**: Zoos can raise awareness of endangered species. |

Table 9: An example taken from Debatepedia test set. highlight indicates key phrases that reflect the purpose. The generated summary from QFS-DLM is almost the same as the gold summary.

**GPT-4 Evaluation**: We rank the summaries of QFS-BART, GENIE, and QFS-DLM using GPT-4 on the Debatepedia and PubMedQA datasets. The results are shown in Table 4. On Debatepedia, our model falls slightly behind QFS-BART in terms of fluency, which may be attributed to the non-autoregressive approach leading to certain limitations in language fluency. However, on PubMedQA, our model outperforms QFS-BART in all metrics, indicating that our model has an advantage over QFS-BART in generating longer summaries. This advantage is attributed to the non-autoregressive approach, which effectively addresses long-term dependencies and exposure bias issues. Furthermore, the summaries generated by our model outperform GENIE in terms of all metrics, illustrating that incorporating query-document relevance in the diffusion language model positively impacts QFS.

**Human evaluation:** As shown in Table 7, we can see that our model performs slightly better than the comparison methods on the Debatepedia and PubMedQA datasets. The improvement in the query-focusing metrics indicates that our approach can better cover the information required by the query.

### 4.6 Case Study

Case studies are conducted to understand the model's performance better. As shown in Table 9, summaries are generated by the proposed method and compared methods for an example chosen from the Debatepedia test set. Although QFS-BART's summary includes the document's content, it cannot answer the query. ChatGPT's summary is too long. In contrast, our summary expresses important information from the document and answers the query like the gold summary.

### 4.7 Query-Document Global Relevance and Sample Quality

Regarding whether the performance of query-document global relevance can reflect the quality of samples, we extracted the top 100 and bottom 100 samples based on query-document global relevance scores from the training sets of Debatepedia and PubMedQA datasets, and named the subsets as "high-relevance" and "low-relevance", respectively.

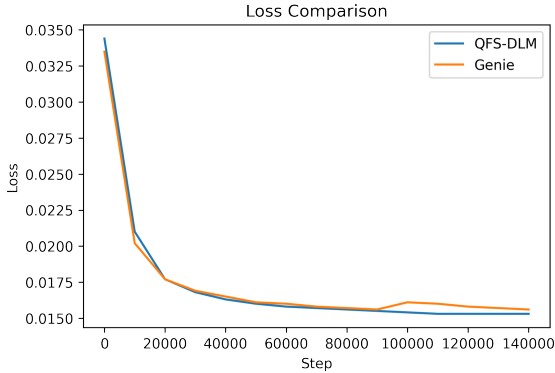

Figure 2: Convergence speed of QFS-DLM and GENIE.

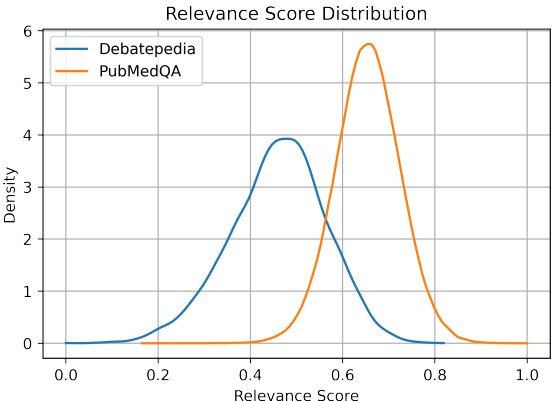

Figure 3: Kernel Density Estimation of Query-Document Relevance Score.

Three text summarization researchers were asked to classify the sample quality as "high-quality" or "low-quality", and the results shown in Table 8, which query-document global relevance could indeed reflect the quality of samples. This finding not only provides a qualitative analysis but also quantitative experimental support.

### 4.8 Impact on Convergence Speed

We performed experiments on the Debatepedia dataset, collected the loss iteration data of the GENIE and our QFS-DLM models, and plotted their convergence curves. As shown in Table 2, our model converges faster.

Both models showed a relatively stable convergence process. However, when GENIE was close to complete convergence, the impact of low-quality data on convergence became significant, resulting in considerable oscillations. On the other hand, QFS-DLM reduced the effect of low-quality data and was able to converge continuously without producing considerable oscillations, resulting in a quicker convergence rate.

### 4.9 Query-Document Global Relevance Analysis

We calculate the global relevance score $R$ between the query and the document. Let $R_{max1}$ denote the highest score and $R_{min1}$ denote the lowest score for Debatepedia, and let $R_{max2}$ denote the highest score and $R_{min2}$ denote the lowest score for PubMedQA. We then normalize the scores using the following equation:

$$R = \frac{R - \min(R_{min1}, R_{min2})}{\max(R_{max1}, R_{max2}) - \min(R_{min1}, R_{min2})} \quad (12)$$

After normalization, we utilize the scipy library [4] to compute the kernel density estimation of the normalized scores, and the results are illustrated in Figure 3. It can be observed that the query-document global relevance scores in the PubMedQA dataset are higher. This disparity could be one reason for the greater improvement observed in PubMedQA compared to Debatepedia when the model incorporates query-document global relevance.

## 5 Conclusion

In this work, we propose QFS-DLM, a diffusion language model that incorporates query-document relevance. Firstly, we extract key fragments from documents based on queries and assign higher weights to them, thereby emphasizing crucial and continuous information within the document. Then, we calculate global relevance scores between queries and documents, and then integrate them into the model's loss function, enabling the model to prefer high-quality data and distance itself from low-quality data. This approach generates summaries that are more focused on query-related information, improving the performance and adaptability of the diffusion language model for the QFS task. Experimental results demonstrate that our model achieves state-of-the-art performance on the Debatepedia and PubMedQA datasets.

### Limitations

Transformer-based summarization models, including diffusion language models, may encounter challenges in efficiently processing long documents. Long documents often contain a substantial amount of information, which makes it difficult to generate concise and accurate summaries while maintaining

---

[4] https://pypi.org/project/scipy/

the focus on the query. Furthermore, the training and testing time of diffusion language models far exceeds that of the BART model, which also restricts its development. Addressing these limitations and conducting further research can lead to a more comprehensive and robust solution.

## Acknowledgements

The authors would like to thank colleagues Qi Lv and Jun Gao for the discussions. The work described in this paper was supported by the National Natural Science Foundation of China (NSFC 62106165).

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
