# OpenReview forum: "Diffusion Language Model with Query-Document Relevance for Query-Focused Summarization"
_EMNLP/2023/Conference — EMNLP 2023 Findings_

### Official Review · Reviewer_SFJ4 · 2023-08-04

**Soundness:** 4

**Excitement:**

3: Ambivalent: It has merits (e.g., it reports state-of-the-art results, the idea is nice), but there are key weaknesses (e.g., it describes incremental work), and it can significantly benefit from another round of revision. However, I won't object to accepting it if my co-reviewers champion it.

**Paper Topic And Main Contributions:**

The paper addresses the problem of Query-Focused Summarisation (QFS). Mainstream QFS models are autoregressive in nature and, thus, suffer from the issues of long term dependencies and exposure bias. Furthermore, low quality data - with lower relevance between queries and documents - can impair model performance.
This paper addresses these two limitations through their main contributions listed below:
1. They apply a non-autoregressive diffusion language model for the QFS task, thus solving the problem of long term dependencies and exposure bias.
2. They incorporate query-document relevance into the loss function of the diffusion language model to enable the model to prefer high-quality data and to adapt well to the QFS task.
3. They conduct experiments on the Debatepedia and PubMedQA datasets, achieving state-of-the-art performance in terms of ROUGE scores and demonstrate promising results in the GPT-4 evaluation.

**Questions For The Authors:**

A. Will integrating query-document relevance into the diffusion model's loss function affect the loss's overall ability to converge? Why or why not?

**Reasons To Accept:**

The main reasons to accept this paper are the following:
1. The paper claims to achieve state-of-the-art performance on the very relevant task of Query-Focused Summarisation, which could have direct implications for improved performance in the task of Question-Answering, which is an area of large-scale interest and efforts. The summaries generated by the new model seem to be of an appropriate, not-too-large length, and more accurate and smoother than the outputs of other compared models.
2. The authors propose a method for the model to intrinsically prefer high-quality data and reject low-quality data during training, by incorporating query-document relevance into the loss function, which could be applied to other Natural Language tasks and reduce the overhead of cleaning curated data before the training processes.
3. The paper provides appropriate mathematical background and explanations for diffusion models in the NLP domain - which is a relatively new area of widespread research, as well as for their specific QFS model.

**Reasons To Reject:**

1. The lack of in-depth analysis and studies to fully understand and explain the effects of having query-document relevance incorporated into the diffusion model's loss function which makes it unclear as to how well the model would generalise and adapt to other domains and datasets (such as facts-based, financial, quantitative, scientific datasets etc.), and to different kinds of queries.
2. The work mostly seems incremental in nature.

**Reproducibility:**

5: Could easily reproduce the results.

**Reviewer Confidence:**

3: Pretty sure, but there's a chance I missed something. Although I have a good feel for this area in general, I did not carefully check the paper's details, e.g., the math, experimental design, or novelty.

---

> ### Author Rebuttal · Authors · 2023-08-29
>
> We sincerely thank you for your conscientious and insightful comments so that we can further refine our paper. We will respond to your comments in the following two aspects:
> ### **1. Incremental work**
>
> Thank you for mentioning this important question.
> In general, **we adapted the diffusion language model for QFS, not a direct application**, and we have provided a description of this in the Introduction section of our paper.
> Specifically, the original diffusion language model lacked sensitivity to queries.
> Furthermore, the quality of the QFS data varies, where query-document relevance can reflect sample quality.
> These two motivations inspired us to modify the original model by incorporating query-document relevance.
> We calculate the query-document relevance score and assign higher/lower loss weights to high-quality/low-quality data, enhancing the  positive impact of high-quality data and reducing the impairment of low-quality data, which guides the model to prefer samples with high query-document relevance to capture query-required messages.
> Experimental results demonstrate that our approach improves summarization performance.
> Additionally, over the past few days, we conducted human evaluations, confirming that samples with higher query-document relevance indeed exhibit better quality, and we will supplement this experiment in the final version of our paper.
>
> ### **2. The effects of incorporating query-document relevance into the diffusion model's loss function**
>
> Thank you for raising such an important issue to further enhance our model.
> In order to better understand the characteristics of our model, we conducted two additional experiments:
> * In the first experiment, we conducted human evaluations on summary quality. **The evaluation showed that our model outperforms the comparison methods across all metrics, including query focusing, fluency, faithfulness, coverage, and overall performance.**
> It is because we enhance the positive impact of high query-relevance data and reduce the impairment of low query-relevance data, allowing the model  to be more biased towards learning from high-quality data.
> We will supplement this experiment in the final version of the paper.
>
> * In the second experiment, we evaluated the convergence speed of the model and found that **our method converged faster than GENIE**.
> The experimental results show that GENIE is easily disturbed by noisy data, which delays the convergence.  Meanwhile, with the help of query relevance, our model is equipped with more refined and effective information, resulting in a smaller impairment of noisy data and a faster convergence rate than GENIE.
> We will provide a detailed description of this experiment below.
>
> We will detailed description of the experiment regarding the loss's overall ability to converge. We performed experiments on the Debatepedia and PubMedQA datasets, collected the loss iteration data of the GENIE and our QFS-DLM models, and plotted their convergence curves.
> The curves showed that our model converges faster.
>
> Below is a brief description of the curves of loss convergence for the Debatepedia dataset:
> * GENIE: The loss value starts at 0.0335 and quickly decreases to 0.0177 within the first 20,000 steps. It continues to decrease slightly to 0.0161 between 20,000-90,000 steps. Between 90,000-100,000 steps, the loss increases slightly to 0.0166 but then slowly decreases again to converge to a final value of 0.0156 between 100,000-140,000 steps.
> * QFS-DLM: The loss value starts at 0.0344 and quickly decreases to 0.0175 within the first 20,000 steps. It then continues to decrease slightly to converge to a final value of 0.0153 between 20,000-100,000 steps.
>
> The convergence of the model on PubMedQA is similar to Debatepedia.
> Both models showed a relatively stable convergence process.
> However, when GENIE was close to complete convergence, the impact of low-quality data on convergence became significant, resulting in considerable oscillations.
> On the other hand, QFS-DLM reduced the effect of low-quality data and was able to converge continuously without producing considerable oscillations, resulting in a quicker convergence rate.
> We will supplement the text content and charts for this experiment in the final version of the paper.

---

### Official Review · Reviewer_ds5i · 2023-08-08

**Soundness:** 4

**Excitement:**

4: Strong: This paper deepens the understanding of some phenomenon or lowers the barriers to an existing research direction.

**Paper Topic And Main Contributions:**

In this paper, the authors develop a new model to address the existing issues in query-focused summarisation. In IR, query-focused summarisation helps obtain the key summary from the relevant documents that could help answer the information sought in the query. As a result, this not only helps with getting answers to the queries quickly but also helps find relevant documents that answer the query.

The authors have noted that the BART model and its variants have been popularly used in query-focused summarisation. The authors mention that the issue is that the models that rely on auto-regressive models face issues with long-term dependencies and bias.

The authors have identified that diffusion models have achieved positive performance that could help address the limitations of the existing methods. To this end, the authors have developed a diffusion language model that incorporates query-document relevance to generate reliable summaries.

**Reasons To Accept:**

The paper addresses the shortcomings in the existing methods in an interesting way by incorporating diffusion models.
The paper compares its models with comparative models in different datasets.
Qualitative and quantitative results demonstrate that the method improves upon existing methods.

**Reasons To Reject:**

Overall, in terms of novelty, this model exploits diffusion models for text replacing standard auto-regressive models. Still, this is not the main criticism of this paper.
One way this paper could be improved is by conducting human experiments. The key question is: do humans understand these summaries?

**Reproducibility:**

4: Could mostly reproduce the results, but there may be some variation because of sample variance or minor variations in their interpretation of the protocol or method.

**Reviewer Confidence:**

3: Pretty sure, but there's a chance I missed something. Although I have a good feel for this area in general, I did not carefully check the paper's details, e.g., the math, experimental design, or novelty.

---

> ### Author Rebuttal · Authors · 2023-08-29
>
> We wish to convey our deepest appreciation to you with wholehearted gratitude.
> Thanks to your thoroughly considered and insightful suggestions, we are able to further polish our paper by conducting necessary experiments and clarifying unclear aspects, which significantly improve the credibility, logic, and structure of our paper.
>
> Thank you for reminding us of the significance of human evaluation.
> As a consequence, we have conducted a human evaluation and the results showed that **our summaries were comprehensible to humans.**
> Specifically, we asked five text summarization researchers to  score the summaries from the perspective of query-focusing (covering the required key information of the query), fluency, faithfulness, coverage, and overall.
> Each perspective was scored on a scale of 1 to 5 points, with a higher score indicating better performance.
>
> As shown in the table below, the results indicate that our model outperforms the comparison methods across all metrics.
> This is because our iter-NAR diffusion model can effectively reduce loss decomposition compared to fully-NAR based on BART variants.
> With enough iterations, the generated summaries become sufficiently stable, and our model is equipped with more refined and effective information, resulting in more fluent summary than GENIE.
> #### Debatepedia dataset (Using 1000 samples from the test set):
> | Summary Sources | Query-Focusing | Fluency | Faithfulness | Coverage | Overall |
> |-----------------|----------------|---------|--------------|----------|---------|
> | QFS-BART        | 3.52           | 4.23    | 3.50         | 3.67     | 3.59    |
> | GENIE           | 3.41           | 4.14    | 3.56         | 3.78     | 3.71    |
> | QFS-DLM         | 4.07           | 4.53    | 4.09         | 4.16     | 4.13    |
>
> #### PubMedQA dataset (Randomly selecting 1000 samples from the test set):
>
> | Summary Sources | Query-Focusing | Fluency | Faithfulness | Coverage | Overall |
> |-----------------|----------------|---------|--------------|----------|---------|
> | QFS-BART        | 3.12           | 4.03    | 3.27         | 3.24     | 3.28    |
> | GENIE           | 2.95           | 3.85    | 3.46         | 3.47     | 3.54    |
> | QFS-DLM         | 3.68           | 4.21    | 3.94         | 4.01     | 3.87    |

---

### Official Review · Reviewer_Feux · 2023-08-12

**Soundness:** 2

**Excitement:**

3: Ambivalent: It has merits (e.g., it reports state-of-the-art results, the idea is nice), but there are key weaknesses (e.g., it describes incremental work), and it can significantly benefit from another round of revision. However, I won't object to accepting it if my co-reviewers champion it.

**Paper Topic And Main Contributions:**

This paper proposes QFS-DLM, a non-autoregressive diffusion language model for query-focused summarization that incorporates query-document relevance and avoids the issues of long-term dependencies and exposure bias that are inherent to autoregressive LLMs, such as BART.

**Reasons To Accept:**

* the proposed method allows incorporating query-document relevance into the diffusion language model, which improves its performance
* the proposed method outperform SOTA approaches

**Reasons To Reject:**

* incremental work: the proposed method largely adopts GENIE, a recently proposed diffusion language model
* limited reproducibility: the code of the proposed method is not released. Coupled with loose definitions of many parameters, this  significantly limits reproducibility
* limited analysis of results: despite superior performance, authors perform little analysis to identify its reasons. Qualitative analysis is boiled down to anecdotal examples, many of which are confusing.

**Reproducibility:**

2: Would be hard pressed to reproduce the results. The contribution depends on data that are simply not available outside the author's institution or consortium; not enough details are provided.

**Reviewer Confidence:**

4: Quite sure. I tried to check the important points carefully. It's unlikely, though conceivable, that I missed something that should affect my ratings.

---

> ### Author Rebuttal · Authors · 2023-08-29
>
> We express our sincere appreciation for dedicating time from your busy schedule to assess our paper and offer multiple constructive insights meticulously.
> We will respond to your comment in the following three aspects:
>
> ### **1. incremental work**
> Thank you for mentioning this important question.
> In general, **we adapted the diffusion language model for QFS, not a direct application**, and we have provided a description of this in the Introduction section of our paper.
> Specifically, the original diffusion language model lacked sensitivity to queries.
> Furthermore, the quality of the QFS data varies, where query-document relevance can reflect sample quality.
> These two motivations inspired us to modify the original model by incorporating query-document relevance.
> We calculate the query-document relevance score and assign higher/lower loss weights to high-quality/low-quality data, enhancing the  positive impact of high-quality data and reducing the impairment of low-quality data, which guides the model to prefer samples with high query-document relevance to capture query-required messages.
> Experimental results demonstrate that our approach improves summarization performance.
> Additionally, over the past few days, we conducted human evaluations, confirming that samples with higher query-document relevance indeed exhibit better quality.
>
> ### **2. limited reproducibility**
> We appreciate your pointing out the re-clarification need for parameter definition and code release.
> Considering that, we will use a symbol table to describe the meaning of each parameter definition and release our code in the paper's final version.
> For the issue of loose definitions of parameters, we found that $\alpha_{t}$ and $\alpha_{i}$  are somewhat confusing, and we rename $\alpha_{i}$ parameter as $R_{i}$.
> After the revisions, our parameter table is presented below.
> Symbols | Description
> --------|------------
> $t$       | time step
> $\beta_{t}$     | $\beta_{t}$ ∈ (0, 1) as different variance scales
> $x_{t}$      | continuous latent representation of the gold summary
> $H_{s}$     | source text representation
> $\alpha_{t}$    |  $\alpha_{t} = 1 - \beta_{t}$
> $R_{i}$      | relevance score of the $i$-th sample between the query and the document
> $Emb(y)$ | embedding of the gold summary
> $u^{t-1}_{\theta}$ | predicted noise
> $\hat{\mu}_{t-1}$ | actual noise
> $z_{\theta}$   | output of the denoising architecture
> $I$       | covariance matrix of Gaussian distribution
>
> ### **3. limited analysis of results**
> Thank you for pointing out this drawback.
> We will revise the experimental results analysis section and add more experiments to demonstrate the effectiveness of our method, and provide a more detailed explanation for the examples in the paper.
>
> In order to better understand our model results and conduct a more in-depth qualitative analysis, the following content is mainly divided into three aspects: Firstly, we briefly explain the experimental results.
> Secondly, we supplement with quantitative experiments to demonstrate that **query-document relevance can indeed reflect sample quality.**
> Finally, we use human evaluation to illustrate the characteristics of our model summaries, and the results show that **our approach outperforms the comparison methods across all metrics.**
>
> #### **3.1 Explanation of the experimental results**
>
> (1) compared with BART variants, QFS-DLM demonstrated more significant results on the long summary dataset.
> This advantage can be attributed to our non-autoregressive approach, which effectively addresses issues related to long-term dependencies and exposure bias.
> (2) By leveraging query relevance, our model is equipped with more refined and effective information, leading to the generation of superior summaries compared to GENIE.
>
> #### **3.2 Correlation between query-document relevance and sample quality**
>
> Regarding whether the performance of query-document relevance can reflect the quality of samples, we extracted the top 100 and bottom 100 samples based on query-document relevance scores from the training sets of Debatepedia and PubMedQA, and named the subsets as "high-relevance" and "low-relevance", respectively.
> Three  text summarization researchers were asked to classify the sample quality as "high-quality" or "low-quality", and the results showed below that query-document relevance could indeed reflect the quality of samples.
> This finding not only provides a qualitative analysis but also quantitative experimental support.
>
> #### Debatepedia dataset:
>
> |                | high-quality | low-quality |
> |----------------|--------------|-------------|
> | high-relevance sample | 89.3%        | 10.7%       |
> | low-relevance sample  | 37.0%        | 63.0%       |
>
> #### PubMedQA dataset:
>
> |                | high-quality| low-quality|
> |----------------|-------|-------|
> | high-relevance sample | 92.3% | 7.7%  |
> | low-relevance sample  | 53.7% | 46.3% |
>
> #### **3.3 Human evaluation**
> In order to obtain statistically significant information, we have conducted a human evaluation in the past few days.
>
> The results are shown in the table below (maximum score of 5 points).
> From the evaluation results, we can see that our model performs better than the comparison methods on the Debatepedia and PubMedQA datasets.  It is because we enhance the positive impact of high query-relevance data and reduce the impairment of low query-relevance data, allowing the model more biased towards learning from high-quality data.
>
> #### Debatepedia dataset (Using 1000 samples from the test set):
>
> | Summary Sources | Query-Focusing | Fluency | Faithfulness | Coverage | Overall |
> |-----------------|----------------|---------|--------------|----------|---------|
> | QFS-BART        | 3.52           | 4.23    | 3.50         | 3.67     | 3.59    |
> | GENIE           | 3.41           | 4.14    | 3.56         | 3.78     | 3.71    |
> | QFS-DLM         | 4.07           | 4.53    | 4.09         | 4.16     | 4.13    |
>
> #### PubMedQA dataset (Randomly selecting 1000 samples):
>
> | Summary Sources | Query-Focusing | Fluency | Faithfulness | Coverage | Overall |
> |-----------------|----------------|---------|--------------|----------|---------|
> | QFS-BART        | 3.12           | 4.03    | 3.27         | 3.24     | 3.28    |
> | GENIE           | 2.95           | 3.85    | 3.46         | 3.47     | 3.54    |
> | QFS-DLM         | 3.68           | 4.21    | 3.94         | 4.01     | 3.87    |

---

### Meta-Review · Area_Chair_3QMC · 2023-09-16

**Recommendation:** 4

**Metareview:**

This paper is about query focused summarization (QFS). It adapts GENIE, a diffusion LM, to query focused summarization. The reviewers highlight that the proposed solution is effective and that it addresses shortcomings of existing autoregressive models via diffusion process that comes with GENIE. Most of the critique of the reviewers centers around the incremental nature of adapting GENIE to QFS, and whether that constitutes a significant contribution. While this is a central point of critique, most reviewers don't rate this as major issue regarding soundness. The next iteration of this paper should, more clearly, elaborate what the novel aspects of this adaptation are and why this is relevant for QFS. Moreover, the reviewers critisize limited analyses, regarding the quality of generated summaries and regarding the reasons for performance improvements. The authors provide additional insights in their response which would benefit the next iteration of this paper. Finally, this paper has been submitted to the efficiency track, though it seems unrelated to that topic. If the authors believe efficiency plays a major role in this submission, they should highlight this more clearly.

---

### Decision · Program_Chairs · 2023-10-07

**Decision:**

Accept-Findings

**Comment:**

This paper is about query focused summarization (QFS). It adapts GENIE, a diffusion LM, to query focused summarization. The reviewers highlight that the proposed solution is effective and that it addresses shortcomings of existing autoregressive models via diffusion process that comes with GENIE. Most of the critique of the reviewers centers around the incremental nature of adapting GENIE to QFS, and whether that constitutes a significant contribution. While this is a central point of critique, most reviewers don't rate this as major issue regarding soundness. The next iteration of this paper should, more clearly, elaborate what the novel aspects of this adaptation are and why this is relevant for QFS. Moreover, the reviewers critisize limited analyses, regarding the quality of generated summaries and regarding the reasons for performance improvements. The authors provide additional insights in their response which would benefit the next iteration of this paper. Finally, this paper has been submitted to the efficiency track, though it seems unrelated to that topic. If the authors believe efficiency plays a major role in this submission, they should highlight this more clearly.